# The Design of a Patient-Centered Hierarchal Composite Outcome for a Multi-Center Randomized Controlled Trial in Metastatic Bone Disease

**DOI:** 10.3390/curroncol32060318

**Published:** 2025-05-30

**Authors:** Hadia Farrukh, Abbey Kunzli, Olivia Virag, Nathan O’Hara, Sheila Sprague, Amy Cizik, Ricardo Gehrke-Becker, Thomas Schubert, Michelle Ghert

**Affiliations:** 1Department of Surgery, McMaster University, Hamilton, ON L8S 4K1, Canada; farrukhh@mcmaster.ca (H.F.); kunzlia@mcmaster.ca (A.K.); viragoe@mcmaster.ca (O.V.); sprags@mcmaster.ca (S.S.); 2Department of Orthopaedics, University of Maryland, Baltimore, MD 21201, USA; 3Department of Orthopaedics, University of Utah, Salt Lake City, UT 84108, USA; amy.cizik@hsc.utah.edu; 4Hospital de Clínicas de Porto Alegre, Porto Alegre 90035-903, Brazil; rbecker@hcpa.edu.br; 5Department of Orthopedic Surgery, Cliniques Universitaires Saint-Luc, 1200 Brussels, Belgium

**Keywords:** metastatic bone disease, proximal femur, randomized controlled trial, resection, reconstruction, internal fixation

## Abstract

The proximal femur represents the most frequent site in the appendicular skeleton for metastatic bone disease (MBD) to occur, with a high risk for pathologic fracture. While surgical stabilization is typically used to manage fractures, reconstruction approaches are gaining popularity due to improved survival. Previous studies have focused on clinical outcomes, but patient-centered outcomes remain underexplored. This study aims to develop a patient-centered primary outcome for the Proximal FEmur Reconstruction or Internal Fixation fOR Metastases (PERFORM) Randomized Controlled Trial, employing a mixed-methods approach. First, a focus group with advanced cancer patients and caregivers identified relevant outcomes. Next, a discrete choice experiment (DCE) assessed the importance of these outcomes among stakeholders, including surgeons, patients and caregivers. The most important components for the primary outcome were identified: mortality within twelve months, physical function assessed at four months using the PROMIS^®^ Global Physical Function score, and the number of days at home within twelve months. The DCE further confirmed that survival and physical function were most prioritized. The PERFORM trial’s primary outcome, developed through extensive stakeholder engagement, will guide the evaluation of surgical approaches for MBD of the proximal femur and has the potential to influence patient-centered practice.

## 1. Introduction

One of the most common locations for cancer to metastasize is the skeleton, known as metastatic bone disease (MBD), or skeletal metastases. Data have shown a steady decline in the cancer mortality rate since 1991, yet a growing prevalence of MBD [1]. An MBD international trend analysis found that the median survival duration after treatment of MBD improved exponentially from 2000 to 2022 [2]. The life expectancy for MBD patients has more than tripled in the last quarter of a century [3]. It is estimated that there are over 300,000 individuals presently living with MBD in North America alone [4].

Among cancer patients with MBD, the proximal femur represents the most common location in the appendicular skeleton and a source of substantial morbidity [5]. The proximal femur can also be subjected to high mechanical forces in the body, making it highly susceptible to cancer-induced bone fragility and subsequent fracture. Surgical stabilization is therefore utilized to protect the stability of the proximal femur, relieve pain, and further improve the patient’s quality of life through maintaining or regaining physical function. Surgical treatment for MBD in the proximal femur has historically involved the stabilization of the pathologic bone with internal fixation, a low cost and low risk technique [3]. However, an alternative, more invasive approach involving resection and reconstruction, is gaining in popularity due to longer life expectancies, lower disease recurrence, and more durable reconstruction in patients with MBD [6].

Previous retrospective studies have focused on the surgical outcomes for these procedures, such as perioperative complications and local tumor recurrence [7,8,9,10]. However, the patient perspective on outcomes, such as quality of life, functional mobility, and living independently, have not been well incorporated into research in this population and, therefore, current evidence may not reflect the patient perspective. Prompted by the need for high level and more patient-centered evidence, the Proximal FEmur Reconstruction or Internal Fixation fOR Metastases (PERFORM) Randomized Controlled Trial (RCT) was designed to answer the following research question: “What is the effect of resection and reconstruction on patient-important outcomes compared to internal fixation for the stabilization of an impending or realized pathologic fracture in patients with MBD of the proximal femur?”.

The comparison of two competing approaches across surgical, functional, and quality-of-life outcomes could lead to a paradigm shift in the treatment of patients with MBD of the proximal femur. However, the ability to conduct an effective and ultimately impactful RCT requires the study to have a well-designed and patient-centered primary outcome. The purpose of this paper is to describe the development of the hierarchal composite primary outcome for the PERFORM trial. We used a mixed methods approach with broad stakeholder engagement. A hierarchical composite outcome was selected because it combines several individual outcomes and offers a more comprehensive and nuanced view of a primary outcome, incorporating both the patient and healthcare provider perspectives in a hierarchal order that is consistent with both perspectives.

## 2. Materials and Methods

Our mixed-methods approach included a two-pronged design in which we first identified patient-centered outcomes by holding a focus group with patients and caregivers with lived experience of advanced cancer. We then assessed the relative importance of these outcomes through a discrete choice experiment (DCE), in which stakeholders completed a survey that was designed to determine what outcomes should be ranked higher, based on their preferences identified through a variety of hypothetical scenarios. Ethics waivers were obtained separately for the focus group and the DCE through the University of Maryland.

### 2.1. Qualitative Research: Focus Group

The research team sought to engage MBD patients and their caregivers in the development of the PERFORM trial primary outcome, specifically in finalizing the hierarchal composite outcome to include patient-important components, and to determine which Patient-Reported Outcome Measures (PROMs) were relevant to the patient population of the PERFORM trial. A PERFORM trial Patient and Caregiver Advisory Group (PCAG) was established. We contacted international colleagues in orthopedic oncology to identify potential patients and caregivers who would be interested in collaborating in the design and conduct of the PERFORM trial. In addition, patient and caregiver partners were recruited through the Canadian Cancer Society. The research team then contacted the members of the PERFORM trial PCAG to gauge their willingness to participate in a focus group discussion regarding their experiences and cancer journeys.

After determining the availability of the 11 members of the PCAG, the research team obtained an ethics waiver from the University of Maryland Institutional Review Board and received written consent from eight patients and caregivers of diverse backgrounds and experiences with advanced cancer. Those who were able to attend participated in a two-hour virtual focus group in August 2024. Participants were provided with a brief demographic questionnaire to complete in advance. The focus group was facilitated by the PERFORM trial research coordinator and a musculoskeletal oncology PROM expert. The focus group guide included the following prompts for discussion:What do you remember most from your (or your family member’s) cancer experience?What component is important for you to maintain a good quality of life, or what impacts your quality of life the most?Has your treatment improved your satisfaction with your daily life?Did you have any complications from surgery or other treatment, and if so, how did that affect your quality of life, satisfaction, and/or anxiety?How would you feel about being randomized to one of two surgical procedures?Which of the following components would indicate to you that your surgery was successful:
a.Quality of life (level of anxiety or satisfaction)b.Mobility or painc.Recurrenced.Re-operatione.Treatment

The meeting was recorded via Zoom, which provided a complete transcript. The transcript was reviewed by a member of the research team to clarify inconsistencies in wording and to remove identifiers. The research team then performed a guided thematic and content analysis to identify the outcomes that the patients/caregivers indicated are important to them and therefore should be included in the hierarchal composite outcome for the PERFORM trial. The analysis also informed the team in selecting the most appropriate instrument to assess PROMs. The transcript was first analyzed separately by each of the facilitators using Dedoose Version 9.1.12 software, and codes were generated. Then we used thematic analysis triangulation to strengthen the findings by comparing the data in the transcript to determine common themes and identify any outliers. A coding structure was then generated based on commonalities, and important quotes were highlighted. The most common themes that emerged were considered priorities for the hierarchal composite outcome for the PERFORM trial.

### 2.2. Quantitative Research: Discrete Choice Experiment

DCEs are a quantitative method to estimate preferences under competing scenarios. While originating in the field of economics, the approach is increasingly applied to health research to determine patients’ treatment and outcome preferences. In this study, we used the DCE to estimate preferences for ranking the candidate components of the PERFORM trial’s primary endpoint.

The DCE presents respondents with a series of hypothetical comparisons. The comparisons are described by attributes with various plausible levels. In this study, the attributes were the candidate components of the composite outcome (mortality, physical function, days at home, and cancer recurrence) selected based on the focus group findings and key informant interviews. The plausible levels for each attribute were based on the literature and clinical experience, and are listed in Table 1. Following a review of the available PROMs to measure physical function and support from a PROMs expert, the research team determined that the Patient-Reported Outcomes Measurement Information System (PROMIS^®^) Global Physical Function score would reflect patients’ desired physical function attribute.

This number of attributes and levels yields 81 possible combinations and 3240 two-alternative comparisons. To optimize the orthogonality of the comparisons and minimize respondent burden, we used a partial factorial design, creating 12 choice sets (i.e., two alternative comparisons) with a D-optimal structure in the JMP Pro Version 17 Choice Design platform (Cary, NC, USA). The choice sets were coded into Qualtrics (Provo, UT, USA) with respondent demographic questions, and they were administered electronically.

The participant responses were analyzed by using a multinomial logit model stratifying by surgeon or patient responses. The data used in the model were effects coded, so the model coefficients were centered on zero and comparable between the attributes. The coefficients indicate the strength and relative direction of the preferences on an absolute scale and are interpreted as the marginal utility derived from the given attribute level. The relative importance of each attribute was calculated by the maximum coefficient value minus the minimum coefficient value in each attribute, divided by the sum of these differences for all attributes. The standard errors were clustered by the respondent. Statistical significance was indicated at a *p*-value of less than 0.05. All statistical analyses were performed using JMP Pro Version 17 (Cary, NC, USA) and R Version 4.4.2 (R Foundation for Statistical Computing).

The DCE was completed electronically by study investigators and their institutional colleagues and by members of the PCAG. The target sample size was based on the formula proposed by de Bekker-Grob et al., which multiplies 500 by the number of levels per attribute (*n* = 3) divided by the product of the number of choice sets (*n* = 12) and the alternatives in each choice set (*n* = 2) [11]. Based on this heuristic, the DCE required at least 63 respondents for sufficient statistical precision in estimating main effects.

## 3. Results

### 3.1. Qualitative Research—Focus Group

The focus group consisted of eight participants, which included people with advanced cancer (*n* = 4) and caregivers (*n* = 4) (Table 2).

The main themes that emerged from the focus group analysis were the following: goals of care and treatment, the impact of cancer treatment on quality of life, and experiences with the healthcare system (Table 3). During the focus group, participants were eager to explain their personal cancer journeys and the experiences of loved ones who they cared for. When recounting their stories, participants explained what was important to them in all settings of care. Factors identified included the following: inpatient clinical experiences and surgical treatments; outpatient clinical experiences such as radiation therapies, homecare, and pharmaceutical treatments; and caring for or receiving care from family members.

Specific phrases were extracted from the transcript during the analysis. One patient described the importance of being at home during this time as, “I want to stay home. I want to be as independent as possible”. Another patient stated, “Quality of life is very individual and personal… it could mean different things for different people. For myself, quality of life might be more based on freedom and independence whereas [for] someone else it might be longevity and spending time with family”. These thoughts highlight the importance of individual patient preferences in the setting of advanced cancer.

The research team further explored patients and caregivers’ thoughts on clinical trials, specifically RCTs involving a surgical intervention. Participants agreed that even when treatment does not contribute to extended life, it is just as important for them to maintain the ability to live life to its full potential. Regarding randomization, patients felt comfortable with the process as long as their surgeons were transparent and honest with them about the potential treatment allocations.

Table 3 outlines the main themes—those weighted as important—of the healthcare system (allow maximum days at home), quality of life/physical activity, and independence at home. Therefore, it was decided that the following would be priorities for the hierarchal composite outcome for the PERFORM trial: mortality within twelve months (life is considered a priority), days at home within twelve months, physical function as assessed by a PROM at four months, and preventing cancer recurrence.

### 3.2. Quantitative Research—Discrete Choice Experiment

In total, the DCE was completed by 86 participants (78 surgeons and eight patients/caregivers, with some overlap with the participants of the focus group). The demographics of the surgeons, patients, and caregivers who completed the DCE are shown in Table 4.

Survival and physical function in everyday activities were the most important outcomes for both investigators and patients/caregivers (Figure 1). For patients/caregivers, survival was favored (relative importance, 44.5%; standard error [SE], 5.3%) but not statistically more than the ability to perform everyday activities (relative importance, 42.1%; SE, 5.3%). Surgeons placed a greater relative importance on improved physical function (relative importance, 51.5%; SE, 1.7%) over survival (relative importance, 31.2%; SE, 1.6%). Increasing days at home was the third most important attribute among patients/caregivers (relative importance, 8.3%; 2.9%) and surgeons (relative importance, 14.7%; SE, 1.2%). Preventing cancer recurrence was the least important for patients/caregivers (relative importance, 5.1%; SE, 2.3%) and surgeons (relative importance, 2.5%; SE, 0.5%). Thus, it was decided that cancer recurrence would not be included as a component in the final composite outcome. The level-specific utilities that correspond to the relative importance estimates are available in Appendix A
Figure A1.

Based on the qualitative data from the focus group and the quantitative data from the DCE, the hierarchal composite outcome was finalized as follows: (1) mortality as assessed at twelve months, (2) physical function as assessed at four months using the PROMIS^®^ Global Physical Function score, and (3) the number of days at home as determined at twelve months.

## 4. Discussion

The success of a large-scale international RCT requires a strong foundation and in-depth stakeholder engagement in primary outcome design. The PERFORM trial primary outcome was developed using patient and caregiver focus group work and a DCE involving surgeons, patients, and caregivers. The final hierarchal composite outcome is therefore a rigorously designed outcome that reflects a patient-centered approach to comparative effectiveness research in MBD. The multiple levels of stakeholder engagement in this aspect of protocol development for the PERFORM trial would not have been possible without funding specifically for this purpose. Previous literature has highlighted some of the many challenges in the execution of RCTs and the importance of designing a robust yet pragmatic protocol with a valid primary outcome [12,13].

Although the PERFORM trial would be the first in orthopedic oncology to illustrate an in-depth protocol and primary outcome development process, the literature provides several examples of protocol development in other medical disciplines that also benefitted from the R34 planning grant. The Therapeutic Hypothermia After Pediatric Cardiac Arrest (THAPCA) trial published a design and protocol overview based on the seven-year planning process that was required to operationalize their multicenter RCT [14]. The outcome selection process during the THAPCA protocol development is described in an additional publication [15]. The THAPCA protocol was conceptualized, developed, and refined over the course of two planning grants through collaboration with research networks, a cohort study, and multiple protocol development meetings. The decade of work in the design and development of the trial, and the commitment of the THAPCA team demonstrates the importance of protocol development as it certainly led to successful funding awards for the THAPCA trials. Another study described the protocol development for HMU! (HIV Prevention for Methamphetamine Users) and demonstrated the use and analysis of focus groups in study development, whose themes were then integrated into the study protocol, similar to the methods described in developing the PERFORM trial [16]. The protocol development study of HMU! highlights the importance of including the experiences of the impacted patient population, and how their recommendations can best inform the trial design and outcome to achieve the greatest success.

This process for the PERFORM trial has highlighted the advantage in establishing a network through collaboration in protocol development. A wide range of stakeholders were involved in the PERFORM trial primary outcome development including study investigators, methodological experts, patients, and caregivers. It is expected that stakeholders who contribute to the development of the primary outcome will find study results more relevant. While many RCTs may consult experts in the field when developing a primary outcome, few will consult patients in this process, although this is now changing, with funding agencies requiring patient engagement to be a component of research projects [17]. Engaging with patients can lead to increased validity and meaningfulness of results by bridging the gap between health research and patient-centered practice [18]. Future patients may also feel more comfortable participating in clinical trials if they know that patients contributed to the study design.

The patient and caregiver stakeholders that were involved in the development process PCAG were limited in number and in diversity. Both the advanced cancer focus group (*n* = 8) and the group of patients/caregivers that participated in the DCE (*n* = 8) consisted of small sample sizes and were primarily recruited within Canada, which may have reduced opinion diversity and introduced some selection bias. However, eight participants can be sufficient in a DCE when the design is highly efficient, and each participant completes a large number of well-structured choice tasks. The DCE had a large number of scenarios in which the participants were able to provide their preferences. This allowed us to gather enough preference data to detect clear patterns despite the small sample size. Cancer recurrence was consistently the lowest priority, probably because ‘days at home’ and survival more tangibly represent this outcome. While the surgeons that participated were of various international backgrounds, the patient perspective captured in the development process did not consider patient opinions in other countries and should be considered in future studies. Where possible, a larger patient sample size should be used in gathering data for outcome development in similar trials to encourage and inform patient-centered evidence.

Another important factor to consider in an international RCT and within the patient population is their access to healthcare and how it varies among countries. Participants in the focus group briefly discussed their experience accessing healthcare services in both Canada and the United States. However, other geographic locations and healthcare centers might not share the same ease of access to surgical techniques and materials such as surgical implants. The PERFORM trial will require participating clinical sites to have access to the surgical implants needed for both randomized treatment allocations. Futures studies can examine the variation in access among different countries and regions. Finally, the PERFORM trial itself will stratify randomization by clinical site, which will help to assess the outcomes across geographic variability. *A priori* subgroup analyses will also address potential effects of geographic variability.

Although the PERFORM primary outcome development process involved consulting a variety of stakeholders, participating stakeholders were predominately surgeons, and we did not engage other stakeholders such as policymakers, hospital administrators, or representatives from insurance companies. Ideally, all 7 Ps of stakeholder engagement (patients and the public, providers, purchasers, payers, policymakers, product makers, and principal investigators) should be involved in the process, but that can result in challenges with differing and biased opinions and a costly increase in complexity [19].

## 5. Conclusions

In conclusion, the PERFORM trial primary outcome was designed with broad stakeholder engagement and resulted in a comprehensive primary outcome that reflects the patient-centered nature of the trial. The hierarchal composite outcome of (1) mortality as assessed at twelve months, (2) physical function as assessed at four months using the PROMIS^®^ Global Physical Function score, and (3) the number of days at home as determined at twelve months, will allow the study team to effectively determine the best surgical approach to patients with MBD of the proximal femur and may ultimately lead to significant change in clinical practice.

## Figures and Tables

**Figure 1 curroncol-32-00318-f001:**
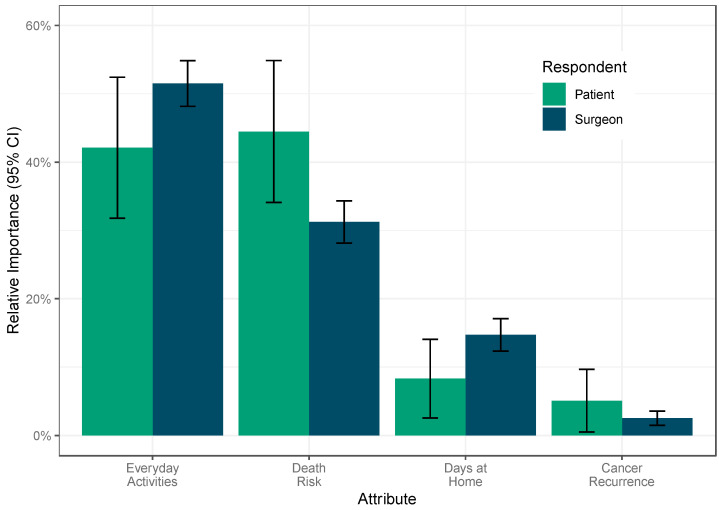
The relative importance of the included attributes stratified by respondent type. CI = confidence interval.

**Table 1 curroncol-32-00318-t001:** Attributes and levels included in the discrete choice experiment.

Attributes	Levels
Days at home (out of hospital) within 12 months of initial surgery	260 out of 365, 300 out of 365, 340 out of 365
12-month risk of death	20%, 40%, 60%
12-month risk of cancer recurrence	5%, 10%, 15%
Ability to carry out every-day physical activities 4 months after initial surgery (i.e., walking, climbing stairs)	Completely, Moderately, Unable

**Table 2 curroncol-32-00318-t002:** Participant characteristics.

Characteristics	Response (*n* = 8)
Age, median in years	55
**Category**	
Patient	4
Caregiver	4
**Self-identified gender**	
Male	1
Female	7
**Education level**	
College/university graduate	7
No answer	1
**Ethnicity**	
White	7
Other	1
**Marital status**	
Married/living with partner	5
Single/never married	2
No answer	1
**Cancer experience**	
Breast	2
Prostate	2
Colon	2
Other	1

**Table 3 curroncol-32-00318-t003:** PERFORM trial focus group thematic structure.

Main Themes	Codes	Sub-Codes
Goals of care and treatment	Medical complications	Side effects and complications
	Satisfaction	
	Education	For patients
	Clinical trials	Benefits/risks
		Randomization
		Comparator group
Quality of life	Mental well-being	Anxiety
		Caregivers
	Physical well-being	Mobility
		Activities of daily living
		Looking good and feeling good
	Individualistic	Needs change for each cancer patient’s individual clinical trajectory
		Overall weakness/holistic wellness
Healthcare system	Interactions with doctors	Impactful conversations
	Continuity	Between healthcare disciplines and teams
		Between different institutions
	Supports available	During active treatment
		At home

**Table 4 curroncol-32-00318-t004:** PERFORM trial discrete choice experiment participant characteristics.

Characteristic	Surgeons *n* = 78	Patients *n* = 8
**Age, median in years (IQR)**	43 (39 to 48)	59 (54 to 64)
**Gender**		
Man	49 (63%)	3 (38%)
Woman	23 (29%)	5 (63%)
Other	2 (3%)	0 (0%)
Prefer not to respond	4 (5%)	0 (0%)
**Race/Ethnicity**		
White	64 (82%)	4 (50%)
Asian	5 (6%)	1 (13%)
African American	1 (1%)	0 (0%)
Latino/Hispanic	9 (12%)	0 (0%)
Other	5 (6%)	1 (13%)
Prefer not to respond	3 (4%)	2 (25%)
**Years in practice, median (IQR)**	10 (5 to 15)	-
**Geographic region**		
North America	47 (60%)	-
Europe	19 (24%)	-
South America	4 (5%)	-
Asia	1 (1%)	-
Prefer not to respond	7 (9%)	-

*n* = number, IQR = interquartile range.

## Data Availability

The original contributions presented in this study are included in the article. Further inquiries can be directed to the corresponding author.

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
