# Peer review of "The Design of a Patient-Centered Hierarchal Composite Outcome for a Multi-Center Randomized Controlled Trial in Metastatic Bone Disease"

_curroncol, 2025, doi:10.3390/curroncol32060318_

Round 1
Reviewer 1 Report
Comments and Suggestions for Authors
This report documents the qualitative and quantitative efforts to select the primary outcome measures for an upcoming RCT on metastatic bone disease. While many of the methods were not familiar, the authors did a commendable job of describing in detail the steps in the process. The outcomes seem quite reasonable for the treatment of this specific pathology.
The only time I felt as if I'd missed an explanation was between the final two paragraphs of 3.1, when the outcomes measures used in the DCE were chosen. Table 3 does not add much value in reaching this important conclusion because there is no weighting to the themes and codes. Given it takes up significant space, I question whether a different representation of the qualitative data could better lead the reader to the specific outcome priorities.
As discussed, the imbalance between the patients and surgeons in the DCE is regrettable given the error bars in Figure 1, and a larger number of patients may have impacted the primary outcome measures, especially cancer recurrence. I assume this would not be submitted as is if obtaining even 8 more patients was easily achievable, so I dont think it needs to postpone publication, but if it could be done, the outcomes would support even greater scrutiny.
Overall very well written and adds future value to the results of the RCT.
line 182: delete "of" after factors.
line 193: add "of", 'importance of individual'
line 196: is --> its
Reviewer 2 Report
Comments and Suggestions for Authors
This manuscript outlines the development of a hierarchical, patient-centered composite primary outcome for the PERFORM trial, which compares proximal femur reconstruction versus internal fixation in patients with metastatic bone disease (MBD). Recognizing that traditional surgical outcomes often exclude patient priorities, the authors implemented a mixed-methods approach, integrating a focus group with patients and caregivers and a Discrete Choice Experiment (DCE) involving both clinicians and stakeholders. The resulting outcome hierarchy prioritizes: Mortality within 12 months; Physical function at 4 months (PROMIS® Global Physical Function score); Days at home within 12 months
The study aims to support a more holistic, patient-relevant endpoint for use in a large international RCT.
This manuscript addresses a relevant and underexplored area in orthopedic oncology—patient-centered outcomes in metastatic bone disease surgery. The mixed-methods approach is sound, and the final composite endpoint is clinically meaningful.
However, before publication, the manuscript requires minor revisions
- While the use of mortality as the top-level outcome is understandable, it would strengthen the rationale to elaborate on how the hierarchy was decided, especially between survival and functional status.
- The focus group and DCE participant samples are small and geographically narrow (primarily Canadian), which may limit generalizability. The authors acknowledge this but could further discuss how this limitation will be addressed in the broader PERFORM trial (e.g., sensitivity analysis by site).
- The authors cite de Bekker-Grob’s rule-of-thumb for DCE sample size, which is acceptable. However, consider adding a power analysis or at least acknowledge the limitations of this heuristic in the context of mixed stakeholder types and low patient sample size.
- The abstract is dense and lacks clear transitions. It could be improved by briefly stating the rationale, methods, main results, and implications in distinct sentences or sections.
- Several minor grammatical errors (e.g., misplaced modifiers, inconsistent verb tenses) should be revised for improved readability.
